# Development of AI-Based Predictive Models for Osteoporosis Diagnosis in Postmenopausal Women from Panoramic Radiographs

**DOI:** 10.3390/jcm14134462

**Published:** 2025-06-23

**Authors:** Francesco Fanelli, Giuseppe Guglielmi, Giuseppe Troiano, Federico Rivara, Giovanni Passeri, Gianluca Prencipe, Khrystyna Zhurakivska, Riccardo Guglielmi, Elena Calciolari

**Affiliations:** 1Department of Clinical and Experimental medicine, University of Foggia, 71122 Foggia, Italy; 2Department of Medicine and Surgery, LUM University, 70010 Casamassima, Italy; 3Department of Medicine and Surgery, Dental School, University of Parma, 43126 Parma, Italy; 4Unit of Medical and Clinical Therapy, Department of Medicine and Surgery, University of Parma, 43126 Parma, Italy; 5Department of Radiology and Nuclear Medicine, Luzerner Kantonsspital, 6000 Lucerne, Switzerland; 6Centre for Oral Clinical Research, Institue of Dentistry, School of Medicine & Dentistry, Queen Mary University of London, London E1 2AD, UK

**Keywords:** osteoporosis diagnosis, panoramic radiographs, postmenopausal women, Dual-energy X-ray Absorptiometry (DXA), radiomics, deep radiomics, convolutional neural networks (CNNs), transfer learning

## Abstract

**Objectives**: The aim of this study was to develop AI-based predictive models to assess the risk of osteoporosis in postmenopausal women using panoramic radiographs (OPTs). **Methods**: A total of 301 panoramic radiographs (OPTs) from postmenopausal women were collected and labeled based on DXA-assessed bone mineral density. Of these, 245 OPTs from the Hospital of San Giovanni Rotondo were used for model training and internal testing, while 56 OPTs from the University of Parma served as an external validation set. A mandibular region of interest (ROI) was defined on each image. Predictive models were developed using classical radiomics, deep radiomics, and convolutional neural networks (CNNs), evaluated based on AUC, accuracy, sensitivity, and specificity. **Results**: Among the tested approaches, classical radiomics showed limited predictive ability (AUC = 0.514), whereas deep radiomics using DenseNet-121 features combined with logistic regression achieved the best performance in this group (AUC = 0.722). For end-to-end CNNs, ResNet-50 using a hybrid feature extraction strategy achieved the highest AUC in external validation (AUC = 0.786), with a sensitivity of 90.5%. While internal testing yielded high performance metrics, external validation revealed reduced generalizability, highlighting the challenges of translating AI models into clinical practice. **Conclusions**: AI-based models show potential for opportunistic osteoporosis screening from OPT images. Although the results are promising, particularly those obtained with deep radiomics and transfer learning strategies, further refinement and validation in larger and more diverse populations are essential before clinical application. These models could support the early, non-invasive identification of at-risk patients, complementing current diagnostic pathways.

## 1. Introduction

Osteoporosis is an age-related musculoskeletal disorder characterized by decreased bone density, heightened bone fragility, and an increased risk of fractures [1]. Early detection and intervention can effectively halt the progression of bone resorption and lower the chance of bone fractures [2]. Osteoporosis attributable fractures (OAFs) are expected to increase from 115,248 in 2010 to 273,794 in 2050, with an estimated total of almost 8.1 million fractures worldwide (78% in women and 22% in men) between 2010 and 2050 [3]. Dual-energy X-ray Absorptiometry (DXA) is considered the gold standard for osteoporosis diagnosis [4,5]. Nevertheless, there is limited predictive power in accurately predicting fractures from BMD measurements [6]. For this reason, DXA is not recommended for general population-based screening; instead, a case-finding strategy is suggested, based on established risk factors or fragility [7].

Radiomics is an emerging field of medical research that focuses on obtaining quantitative metrics or radiomic characteristics from different types of 2D or 3D imaging with the aim of improving diagnostic accuracy and clinical decision-making [8]. Together with other clinical data, these aspects provide important information that may help solve challenging clinical problems. These characteristics include tissue shape, density, texture, and heterogeneity [9]. Through the examination of characteristics seen in radiographic pictures, radiomics can develop accurate prediction models to aid in clinical judgment [10]. In this respect, dental panoramic radiographs (OPTs) show potential for opportunistic osteoporosis screening by visualizing changes in the mandibular bone structure. Various indices and methods applied to OPTs have been tested over the years, producing mixed, but generally promising, results [11,12,13,14]. In particular, studies have demonstrated that radiomorphometric indices, such as the Mental Index (MI), Gonial Index (GI), and Antegonial Index (AI), extracted from OPTs, can be moderately accurate in identifying patients at risk of low bone mineral density [15]. Additionally, combining these indices with patient factors like BMI and age has been explored to enhance predictive power, although with limited sensitivity [16]. Nakamoto et al. also investigated the diagnostic performance of untrained general dental practitioners in visually assessing mandibular cortical erosion from OPTs, revealing that panoramic radiographs could be a useful adjunct in identifying postmenopausal women who should be referred for DXA [17]. With the advances in artificial intelligence applied to imaging, deep learning and machine learning techniques have been explored for osteoporosis diagnosis based on OPT analysis [18]. The end-to-end CNN approach has produced promising results, but it has not been linked to the DXA diagnostic method, which is currently the gold standard for osteoporosis diagnosis [19,20].

This study aims to explore the effectiveness of AI-driven radiomics in diagnosing osteoporosis using OPT images. In recent years, artificial intelligence has increasingly found applications in dentistry, ranging from caries detection and periodontal disease classification to oral cancer screening and orthodontic planning. This trend is part of a broader movement toward the integration of AI for diagnostic support and decision-making in clinical medicine. In particular, convolutional neural networks (CNNs) have shown promising results in analyzing dental radiographs, enabling automated and reproducible assessments of bone and tooth structures that may otherwise require expert interpretation [21,22]. It is hypothesized that various AI methodologies, including classical radiomics, deep radiomics, and convolutional neural networks (CNNs), can effectively differentiate between patients with and without osteoporosis by detecting subtle patterns in jawbone imagery that may not be readily apparent to the human eye. By developing and comparing several predictive models, this research seeks to establish a novel, accessible approach to osteoporosis screening and diagnosis, potentially leading to earlier interventions.

## 2. Materials and Methods

### 2.1. Study Cohort

This study was conducted with a retrospective design including panoramic radiographs (OPTs) from two university cohorts (the University of Parma and the Hospital of San Giovanni Rotondo) in Italy. All the procedures were conducted following the recommendations of the Declaration of Helsinki, as revised in Fortaleza (2013), for investigations involving human subjects, and the STROBE guidelines were also taken into consideration [23]. The study protocol was approved by the Ethical Committee of the University of Parma (No. 34568—1 September 2023, trial registration number NCT03304743) and by the Ethical Committee at the Casa Sollievo della Sofferenza of San Giovanni Rotondo (No. 126/2023) in Italy. To be included in the cohort (Appendix A), patients had to meet the following criteria: women aged 50 years and older with self-reported menopause (defined as the permanent cessation of menstruation for at least one year) who had had a DXA examination of the hip and lumbar spine performed within the previous 12 months. The time interval between DXA assessment and panoramic radiograph acquisition was within three months for all patients included in the study. A T-score ≤ −2.5 at the hip or spine was necessary for inclusion as a subject with osteoporosis, whereas healthy controls required a T-score ≥ −1. Participants were selected based on the availability of high-quality OPT images and a documented clinical diagnosis regarding osteoporosis status. The study specifically included individuals with comprehensive medical and dental records.

Participants were excluded if affected by systemic diseases other than osteoporosis that severely affect bone metabolism, such as Cushing’s syndrome, Addison’s disease, type 1 diabetes mellitus, leukemia, pernicious anemia, malabsorption syndromes, chronic liver disease, or rheumatoid arthritis. Additional exclusion criteria (Appendix A) included a known infection with HIV or viral hepatitis, a history of local radiation therapy within the last five years, limited mental capacity or language skills impeding understanding of study information or consent, and severe acute or chronic medical or psychiatric conditions or laboratory abnormalities that could interfere with trial participation or the interpretation of results.

### 2.2. Imaging Protocol

Each participant recruited at both centers underwent OPT using the ORTHOPHOS XG 3D (Sirona Dental, Charlotte, NC, USA) standardized digital panoramic dental X-ray machines. The OPT images were acquired following standard clinical protocols ensuring diagnostic quality. The images provide a comprehensive view of the lower face, displaying both jaws, all teeth, and the surrounding bone structure.

### 2.3. Definition of ROI and Image Processing

An area that was visible and spread apically to the mandibular canal and distally to the mental foramen, including the angle of the mandible and a portion of the ascending ramus up to the mandibular foramen, was deemed appropriate for inclusion in this study (Figure 1a). This area was subsequently designated as the region of interest (ROI) (Figure 1b). All ROIs were manually traced and cropped from the original OPTs using the “lasso selection” command in the Preview application (Apple Inc., Cupertino, CA, USA). The ROIs were first converted to grayscale and then resized to 256 × 256 pixels to balance image quality and computational efficiency. Histogram equalization was performed on the ROIs to equally distribute the levels of gray intensity with the aim of reducing contrast variations due to possible differences in acquisition protocols. To ensure robust model evaluation and minimize data leakage, the dataset was randomly split at the patient level into training (80%), validation (10%), and test (10%) sets using a reproducible random seed. External validation was conducted on an independent cohort of 56 OPTs from the Parma clinical center. Images with significant artifacts or poor diagnostic quality (motion blur, improper patient positioning) were excluded following a preliminary image quality assessment.

### 2.4. Classical Radiomics

Radiomic features were extracted from the normalized ROIs using Python 3.11.8, using a package called *Pyradiomics*. This framework allows for the extraction of a wide range of radiomic features from images, including those related to shape, texture, and intensity, facilitating detailed analysis of imaging data. The extracted features were then saved in a CSV file, enabling easy aggregation and analysis of radiomic data and the target column indicating the positive or negative diagnosis of osteoporosis for each patient. This step was implemented to subject the structured data to a supervised machine learning workflow. A preprocessing and feature selection process was carried out on the dataset for analytical purposes, using machine learning methods to optimize the input for the predictive models. The “Fast Correlation-Based Filter” (FCBF) function was applied to perform feature selection for further analysis. Various supervised machine learning models were defined, including logistic regression, SVC, Gaussian NB, and Decision Tree. This allowed an evaluation of which model performed best on the selected features. Each model was trained on the training dataset and then evaluated on the test set.

### 2.5. Deep Radiomics

For deep radiomics feature extraction, ROIs were loaded from two different folders: one containing images of normal subjects and the other containing images of subjects with osteoporosis. Using five different CNN models (ResNet50, DenseNet-121, VGG-16, Inception V3, and EfficientNet) deep features were extracted from the ROIs. The extracted features were filtered using a FCBF algorithm to identify the most relevant ones for differentiating between healthy subjects and those with osteoporosis. Subsequently, various machine learning models (logistic regression, Random Forest, SVM, Naive Bayes, K-Nearest Neighbors, Decision Tree, Gradient Boosting) were trained on the training data and evaluated on the test set.

### 2.6. End-to-End CNNs

In this phase, different CNN approaches were applied to predict the onset of osteoporosis using previously selected and normalized OPT ROIs. The images, previously labeled based on the presence or absence of osteoporosis, were randomly assigned among the different sets according to the specified proportions, using a randomization algorithm to ensure a balanced distribution.

Pre-trained ResNet50, DenseNet, VGG-16, Inception V3, and EfficientNet models were used. The aim was to transfer the information previously gained from the ImageNet dataset by pre-trained CNNs to the cohort of patients who were part of this study; the CNNs had previously been trained on large-scale image datasets. The four distinct transfer learning strategies used were Feature Extraction Hybrid, Feature Extraction Non-Hybrid, Fine-Tuning 50%, and Fine-Tuning from Scratch. These models were leveraged to utilize the weights learned from the ImageNet dataset, exploiting transfer learning to enhance performance on a different, possibly smaller dataset. To address the limitation of the modest dataset size, advanced data augmentation techniques were employed during training. These included random rotations (±30°), horizontal and vertical flips, zoom and shear transformations, brightness variation, channel shifting, and translations. Such augmentations enriched the diversity of the training images, effectively simulating real-world variability and reducing the risk of overfitting.

### 2.7. Performance Evaluation

Hyperparameter tuning was performed to optimize the model’s performance. The “Optuna” framework was utilized for the systematic exploration of the hyperparameter space, employing Bayesian optimization techniques to effectively search for optimal settings. Key parameters such as *learning rate* and *batch size* (ranging from 16 to 64) were optimized based on the objective to maximize the AUC. Different trials were conducted to find the best combination of learning rate and batch size that resulted in the highest validation AUC. Furthermore, early stopping was implemented based on the loss curves during training. This process ensured that the model not only fit well but also generalized better to unseen data, thus enhancing its predictive accuracy and reliability. In this study, the performance of all models was evaluated using a comprehensive set of metrics: Area Under the Curve (AUC), accuracy, sensitivity, and specificity. These metrics provide a well-rounded assessment of the models’ capabilities in distinguishing between patients with and without osteoporosis based on orthopantomography (OPT) images. For each approach employed in the study—classical radiomics, deep radiomics, and end-to-end convolutional neural networks (CNNs)—the models were trained on the training set and evaluated on the validation and test sets. Additionally, a final external validation was conducted in which the models derived from the three approaches were tested on an independent dataset to further assess their generalizability and robustness. All the analyses were performed in Python using various packages, including TensorFlow and Keras.

## 3. Results

### 3.1. Participant Cohort

Initially, 597 panoramic radiographs (OPTs) from postmenopausal women were screened across two clinical centers: 145 from the University of Parma and 452 from the Hospital of San Giovanni Rotondo. Among them, 126 subjects were classified as healthy, 296 as osteopenic, and 175 as osteoporotic. To ensure a clearer diagnostic separation, all osteopenic cases were excluded from the analysis. The final study cohort included 301 women: 126 healthy and 175 osteoporotic. Specifically, 245 OPTs (105 healthy, 140 osteoporotic) from San Giovanni Rotondo were used for model training and internal testing, while 56 OPTs (21 healthy, 35 osteoporotic) from Parma served as a separate external validation set. All relevant details are summarized in Table 1.

### 3.2. Classical Radiomics

For each of the 245 patients, 75 different types of radiomic features were extracted. Subsequent feature selection via FBCF led to the selection of four radiomic features (original_firstorder_90Percentile, original_firstorder_Mean, original_firstorder_Range, original_firstorder_RobustMeanAbsoluteDeviation, original_glcm_ClusterShade) that were strongly correlated with the output but weakly correlated with each other, thereby enhancing the diversity of the selected features. Table 2 presents the performance in terms of accuracy and AUC for the various machine learning models tested using the abovementioned features and variables. The best performance in terms of AUC and accuracy was achieved by the logistic regression model with an accuracy of 0.641 and an AUC of 0.514 in external validation. All the results related to classical radiomics, including those from the internal test and external validation, are provided in Appendix A.

### 3.3. Deep Radiomics

For each CNN architecture, a large number of deep features were initially extracted per ROI: 128,000 with EfficientNet, 1000 with ResNet50, 25,088 with VGG16, 82,944 with DenseNet121, and 131,072 with InceptionV3. Subsequently, the seven most relevant features for each model were selected using feature selection techniques based on relevance and redundancy criteria, such as the Fast Correlation-Based Filter (FCBF). For each CNN architecture, logistic regression was identified as the best-performing machine learning model based on AUC, as shown in Appendix A. The external validation results of these models, including AUC, accuracy, sensitivity, and specificity, are reported in Table 3. Among all the models, DenseNet-121 achieved the highest AUC (0.722), followed by EfficientNet-b0 (0.683), InceptionV3 (0.605), ResNet50 (0.610), and VGG16 (0.473).

The corresponding results obtained from the internal test set are provided in Appendix A.

### 3.4. End-to-End CNNs

In the external validation, the performance of the end-to-end CNN models was substantially lower than that in the internal test set, underscoring the persistent challenge of achieving robust generalizability. Across all the transfer learning strategies employed, none of the architectures exceeded an AUC of 0.80, indicating limited discriminative ability on unseen data.

Using the hybrid feature extraction approach, EfficientNet-b0 exhibited markedly poor predictive performance, achieving an accuracy of 0.375, an AUC of 0.437, and a sensitivity of 0%. Despite its perfect specificity (100%), the model’s complete failure to identify positive cases renders it clinically unsuitable. Other models, including VGG16 and ResNet-50, demonstrated slightly improved outcomes, with ResNet-50 attaining an AUC of 0.786 and a sensitivity of 90.5%, albeit with a specificity of only 37.1%.

Under the non-hybrid feature extraction paradigm, several models displayed increased sensitivity, though at the cost of specificity. Notably, DenseNet-121 and ResNet-50 achieved perfect sensitivity (100%) but exhibited very low specificity (0–14.3%). InceptionV3 yielded an AUC of 0.668, coupled with a high sensitivity of 97.1%, yet failed to correctly classify any negative cases.

In the 50% fine-tuning configuration, EfficientNet-b0 demonstrated the most balanced, though still suboptimal, performance, with an accuracy of 0.679, an AUC of 0.680, a sensitivity of 91.4%, and a specificity of 28.6%. Comparable AUCs were obtained with ResNet-50 (0.615) and InceptionV3 (0.652), but both models continued to exhibit poor specificity, remaining below 39%.

Finally, in the full fine-tuning from scratch setting, none of the models achieved an adequate balance between sensitivity and specificity. Although InceptionV3 attained a sensitivity of 97.1%, its specificity plummeted to 0%, while ResNet-50 and DenseNet-121 maintained modest AUCs between 0.567 and 0.615.

A comprehensive summary of performance metrics from the external validation is presented in Table 4, whereas the corresponding internal test results are detailed in Appendix A.

## 4. Discussion

Over the years, efforts to reduce human subjectivity in the diagnostic process have led to studies exploring the use of machine learning techniques such as Support Vector Machines (SVMs), decision trees, and texture analysis (GLCM), to enhance the accuracy of osteoporosis diagnosis [18,24]. One of the main innovations was the decrease in variability related to human diagnosis, which led to an increase in generalization ability.

The present study showed the potential of radiomic and deep learning models in osteoporosis screening from OPTs. In particular, it suggested that the best-performing model in deep radiomics was logistic regression (AUC of 0.722) applied to features extracted using DenseNet-121. Focusing on end-to-end CNN approaches, the best performance in external validation was achieved by ResNet-50 using the hybrid feature extraction strategy, which achieved an AUC of 0.786, a sensitivity of 90.5%, a specificity of 37.1%, and an accuracy of 0.571. This configuration demonstrated a strong ability to correctly identify patients with osteoporosis, minimizing false negatives and making it potentially suitable for opportunistic screening purposes. However, the low specificity indicates a substantial rate of false positives, which could lead to over-referral for DXA in healthy individuals. This trade-off reflects a cautious model behavior, favoring sensitivity over specificity, and may be acceptable in contexts where missing a diagnosis carries greater clinical risk than overdiagnosis. Notably, these results were obtained on an independent external validation set, supporting their robustness beyond the training data. Importantly, the AI models developed in this study were trained to identify patterns in the mandibular bone structure that are not readily visible to the human eye. Classical radiomics captures hand-crafted features such as first-order statistics and texture metrics (GLCM, GLRLM), which quantify bone homogeneity, granularity, and architectural organization parameters that are difficult to assess visually [9]. Deep radiomics leverages pre-trained convolutional neural networks to extract thousands of abstract features from intermediate network layers; these features encode complex spatial hierarchies and subtle morphological cues embedded in the trabecular pattern and cortical outline without relying on human-defined rules. Finally, end-to-end CNNs autonomously learn discriminative representations directly from pixel-level data, optimizing filters across multiple layers to recognize imperceptible differences in bone density, trabecular alignment, cortical thinning, and shape deformations. These automated approaches reveal microstructural signatures of systemic bone fragility that may elude even experienced radiologists during standard panoramic evaluation.

In the past, based on a preliminarily diagnosed osteoporosis using the DXA system, Hwang et al. reported encouraging results in terms of accuracy and AUC using classical machine learning models [18]. Recent research has begun to employ CNNs coupled with transfer learning to analyze panoramic radiographs [19,20]. These studies marked a significant qualitative leap in the approach to diagnosing osteoporosis. CNNs achieved encouraging AUC scores, indicating superior discriminative ability [20]. The advantages of using deep learning include its high capacity to learn complex image representations, improved sensitivity and specificity, and reduced dependency on image preprocessing. The main drawback of the aforementioned studies is that the preliminary diagnosis is made based on radiomorphometric indices, introducing a significant bias into the study design.

Conversely, in the current study, all patients were diagnosed based on DXA assessment. This approach provides greater diagnostic certainty compared to the use of radiomorphometric indices applied to mandibles, which are highly operator-dependent. CNNs, combined with transfer learning, have performed well, yielding results consistent with trends in the scientific literature.

In this work, in addition to radiomics, further image analysis approaches based on various CNN architectures and transfer learning (TL) techniques were utilized. Transfer learning (TL) is based on the concept that knowledge acquired in one context can be applied to another. It is based on the functioning of the human brain [25]. For instance, learning to ride a bicycle makes learning to ride a motorcycle much easier. This concept is also prevalent in learning academic subjects, pattern recognition, and acquiring new languages [26]. This concept has recently been extended into the AI space with the goal of using image recognition expertise for medical image analysis. It is assumed that general visuals and medical images may share useful visual features and that transfer learning can help CNNs assimilate new information [27].

Due to the possibility of incorrect results from the anterior region’s overlapping cervical vertebrae, all of the ROIs for this study were selected from the posterior mandible. Furthermore, the posterior region is less susceptible to blurring than the anterior region due to the thicker focal trough, which is less influenced by the patient’s position [28]. All machine and deep learning models used in this study are widely accessible due to open-source libraries like Tensorflow(2.19.0) and PyTorch(2.7.1). However, training deep learning models can demand significant computational resources, especially for larger datasets or complex architectures. While pre-trained networks reduce the training load, specialized hardware, like GPUs, may still incur additional costs. Despite these considerations, the workflow remains cost-effective due to the availability of open-source software, though the necessary infrastructure for training and testing can be a limiting factor for smaller institutions [29].

The present study also has some limitations that should be considered. CNNs have several layers in their neural network, meaning that they need a large number of images to be trained [30]. It is difficult to find a medical field with such a big database of data. Deep learning systems are typically trained on more than 1000 medical radiological images; however, 245 cases might provide an acceptable outcome with data augmentation and by reusing the pre-trained network [31]. Nonetheless, the relatively small sample size may still pose a risk of overfitting, particularly for more complex CNN architectures. This limitation could compromise the generalizability of the models to new populations and different imaging conditions. Although internal testing yielded high AUC values, including instances of apparent perfect classification (AUC = 1.0), such results are likely indicative of overfitting to the training data rather than true model performance (Appendix A). The more modest outcomes observed during external validation reinforce the need for cautious interpretation and highlight the necessity of rigorous generalizability assessments. Overfitting remains a significant concern, especially when working with limited datasets, and must be carefully addressed through the development of larger, prospective multicenter studies before these AI models can be reliably deployed in clinical practice. As a first step to mitigate this risk, we included an independent external validation cohort in our study; however, further large-scale validation remains necessary. Another limitation of this study is the lack of information regarding the pharmacological treatment status of the participants with osteoporosis at the time of panoramic radiograph acquisition. It remains unclear whether these patients had been newly diagnosed or were already undergoing anti-osteoporotic therapy, which could potentially influence mandibular bone characteristics and, consequently, the radiomic or deep learning features extracted from OPTs. Additionally, the study did not record participants’ ethnicity, although it is known that racial and ethnic background, particularly the distinction between Caucasian and non-Caucasian populations, can significantly influence osteoporosis prevalence and bone density patterns. These aspects should be considered in future multicenter studies with more diverse and well-documented populations.

Despite these limitations, the sample size used in our study aligns with recent literature standards. An a priori sample size estimation was not performed due to the retrospective nature of this study. However, a post hoc evaluation of sample adequacy was conducted. For convolutional neural network (CNN) models, the literature suggests that a minimum of approximately 120–150 subjects per group is needed to achieve reliable results using transfer learning techniques, assuming an expected AUC of 0.80, a two-sided α of 0.05, and a power of 80%. In our study, 245 OPT images were included in the internal training and test cohort. Given that transfer learning techniques were applied, this sample size exceeds the minimum threshold and is consistent with the standards reported in recent CNN-based studies [32].

For classical radiomics and deep radiomics approaches, which rely on feature extraction followed by supervised machine learning rather than full image-based learning, the sample size requirements are generally less demanding. These methods benefit from dimensionality reduction techniques and feature selection algorithms (FCBF), which help mitigate the risk of overfitting when working with moderate-sized datasets. [33,34].

Additionally, to evaluate the generalizability of all models, an external validation set of 56 OPTs from an independent clinical center was used, supporting the robustness and potential clinical utility of our findings.

Another limitation is the absence of comparison between the performance of AI models and that of expert radiologists or conventional radiomorphometric indices such as the mandibular cortical index. While these comparisons could be important for evaluating clinical utility, they were not included in the present retrospective design and are suggested as a key objective for prospective validation studies.

Finally, all the ROIs in this study were manually delineated, introducing a potential operator-dependent bias and limiting reproducibility. Although this approach aligns with the previous literature, the lack of intra- and inter-observer variability analysis further reduces methodological robustness. Future research should aim to incorporate standardized and reproducible semi- or fully-automatic segmentation methods, which could improve objectivity, reduce observer bias, and enhance clinical applicability. However, implementing such automation would require a complete reprocessing of the data pipeline and retraining of the models.

The stringent selection criteria and exclusion considerations ensured that the study population accurately reflected the conditions under investigation, allowing for precise assessments and valid conclusions. This strict selection of clearly defined osteoporotic and healthy controls maintained the integrity and specificity of the study’s findings.

Although the results are encouraging and underscore the potential of radiomics and deep learning for osteoporosis prediction from panoramic radiographs, further refinement and validation on larger, more heterogeneous cohorts are necessary to confirm their clinical applicability. Nevertheless, utilizing ResNet50, the end-to-end CNN method emerges as the most promising for osteoporosis diagnosis through OPTs.

Future developments could explore the capability of AI systems not only to detect osteoporosis but also to identify other jawbone conditions potentially linked to systemic bone metabolism, including medication-related osteonecrosis of the jaw (MRONJ), whether drug-induced or idiopathic. Integrating these additional objectives may broaden the clinical value of AI-driven analysis of dental radiographs [35].

## 5. Conclusions

This study demonstrated the feasibility of using AI-based models, particularly deep radiomics and transfer learning approaches, to predict osteoporosis from panoramic radiographs in postmenopausal women. While classical radiomics demonstrated limited performance, CNN-based models demonstrated promising performance, especially when evaluated on an external validation cohort. Although further validation on larger and more diverse datasets is necessary, these findings support the potential role of panoramic radiographs as a non-invasive, accessible screening tool for early osteoporosis detection in dental settings.

## Figures and Tables

**Figure 1 jcm-14-04462-f001:**
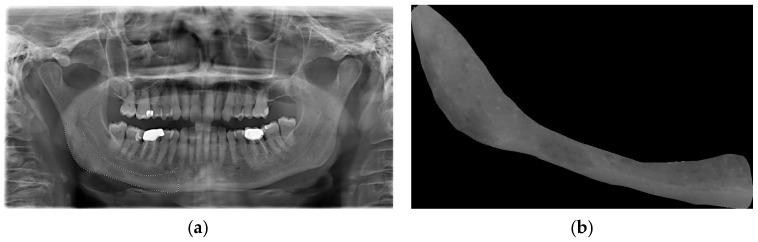
(**a**) OPT used for region-of-interest (ROI) drawing on the right site of the mandible. (**b**) Region of interest cropped from the original OPT and used for subsequent analysis.

**Table 1 jcm-14-04462-t001:** Distribution of the initially collected cases and the final study cohort, categorized by center and diagnosis.

Diagnosis	Total	Parma	San Giovanni Rotondo
Normal	126	21	105
Osteopenia (excluded)	296	89	207
Osteoporosis	175	35	140
Total screened	597	145	452
Final cohort	301	56	245

**Table 2 jcm-14-04462-t002:** Performance in terms of accuracy, AUC, sensitivity, and specificity for the various machine learning models tested for the classic radiomics approach.

	Logistic Regression	SVC	Gaussian NB	Decision Tree
**AUC**	0.5147 [0.5000, 0.5484]	0.3638 [0.2608, 0.4634]	0.5147 [0.5000, 0.5484]	0.5000 [0.5000, 0.5000]
**Accuracy**	0.6415 [0.5094, 0.7736]	0.3585 [0.2264, 0.4906]	0.6415 [0.5094, 0.7736]	0.3585 [0.2264, 0.4906]
**Sensitivity**	1.0000 [1.0000, 1.0000]	0.0000 [0.0000, 0.0000]	1.0000 [1.0000, 1.0000]	0.0000 [0.0000, 0.0000]
**Specificity**	0.0000 [0.0000, 0.0000]	1.0000 [1.0000, 1.0000]	0.0000 [0.0000, 0.0000]	1.0000 [1.0000, 1.0000]

**Table 3 jcm-14-04462-t003:** The best performances of the supervised machine learning models for each type of CNN used in the deep radiomics approach, in terms of AUC, accuracy, sensitivity, and specificity, are reported for the external validation.

	EffiecientNET-b0	ResNET50	VGG16	DenseNet-121	InceptionV3
**AUC**	0.683 (95% CI [0.538, 0.819])	0.610 (95% CI [0.457, 0.754])	0.473 (95% CI [0.302, 0.632])	0.7224 (95% CI [0.5735, 0.8516])	0.605 (95% CI [0.454, 0.755])
**Accuracy**	0.589 (95% CI [0.464, 0.714])	0.536 (95% CI [0.410, 0.661])	0.446 (95% CI [0.321, 0.571])	0.5714 (95% CI [0.4464, 0.7143])	0.589 (95% CI [0.464, 0.714])
**Sensitivity**	0.486 (95% CI [0.322, 0.667])	0.571 (95% CI [0.412, 0.730])	0.457 (95% CI [0.294, 0.618])	0.4286 (95% CI [0.2647, 0.6000])	0.629 (95% CI [0.469, 0.778])
**Specificity**	0.762 (95% CI [0.550, 0.933])	0.476 (95% CI [0.261, 0.706])	0.429 (95% CI [0.211, 0.636])	0.8095 (95% CI [0.6087, 0.9565])	0.524 (95% CI [0.300, 0.739])

**Table 4 jcm-14-04462-t004:** Model performance of the five CNN architectures using four types of transfer learning techniques in external validation.

CNN Architecture	Best Transfer Learning Strategy	AUC (95% CI)	Accuracy (95% CI)	Sensitivity (95% CI)	Specificity (95% CI)
ResNet-50	Hybrid Feature Extraction	0.786 (0.654–0.894)	0.571 (0.446–0.696)	0.905 (0.769–1.000)	0.371 (0.212–0.517)
DenseNet-121	Non-Hybrid	0.727 (0.593–0.852)	0.625 (0.500–0.750)	1.000 (1.000–1.000)	0.000 (0.000–0.000)
InceptionV3	50% Fine-Tuning	0.668 (0.506–0.803)	0.661 (0.536–0.768)	0.829 (0.694–0.944)	0.381 (0.182–0.579)
EfficientNet-b0	50% Fine-Tuning	0.680 (0.518–0.820)	0.679 (0.554–0.786)	0.914 (0.816–1.000)	0.286 (0.105–0.476)
VGG16	Non-Hybrid	0.631 (0.474–0.781)	0.625 (0.500–0.750)	1.000 (1.000–1.000)	0.000 (0.000–0.000)

## Data Availability

Data supporting the findings of this study are available from the corresponding author upon reasonable request.

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
