# Peer review of "Development of AI-Based Predictive Models for Osteoporosis Diagnosis in Postmenopausal Women from Panoramic Radiographs"

_jcm, 2025, doi:10.3390/jcm14134462_

Round 1
Reviewer 1 Report
Comments and Suggestions for Authors
Problems in the Study Design:
- The sample size is small.
- The control group size is not matched with the sample (study) group.
- It is not mentioned whether the participants belong to the Caucasian race, which is an important factor as ethnicity influences osteoporosis incidence.
- There is no information on whether patients with osteopenia were included in the sample.
- No comparison is made between the proposed AI model and the morphometric indices used in studies based on OPT (orthopantomograms), which are known for high sensitivity and specificity.
- In the study cohort description, it is not specified whether patients taking medications (e.g., affecting bone metabolism) were excluded.
- It is not clarified whether the DEXA scan coincided temporally with the OPT acquisition.
- It is not mentioned whether patients were newly diagnosed with osteoporosis or if they had previously received treatment.
- The study does not explain what the AI model identifies that the human eye cannot detect.
- The sample size is too small for reliable use of the Python code developed.
- In the discussion section, the authors draw arbitrary conclusions, despite the known fact that in morphometric studies, the variability related to human diagnosis (intra-observer and inter-observer agreement) is very high.
- There is no supporting literature to substantiate several claims.
- No sample size calculation was performed.
- Finally, the study fails to demonstrate that the AI model is faster or better than the established morphometric indices, and no relevant bibliography regarding the use of morphometric indices in OPT is provided.
Author Response
Problems in the Study Design:
- The sample size is small. The control group size is not matched with the sample (study) group.
The manuscript has been revised to address this issue. A dedicated paragraph has been added to the discussion section to clarify and justify the sample size and the group composition. The following passage has been included:
“Despite these limitations, the sample size used in our study aligns with recent literature standards. A priori sample size estimation was not performed due to the retrospective nature of this study. However, a post-hoc evaluation of sample adequacy was conducted. For convolutional neural network (CNN) models, literature suggests that a minimum of approximately 120–150 subjects per group is needed to achieve reliable results using transfer learning techniques, assuming an expected AUC of 0.80, a two-sided α of 0.05, and a power of 80%. In our study, 245 OPT images were included in the internal training and test cohort. Given that transfer learning techniques were applied, this sample size exceeds the minimum thresholds and is consistent with the standards reported in recent CNN-based studies.
For classical radiomics and deep radiomics approaches, which rely on feature extraction followed by supervised machine learning rather than full image-based learning, the sample size requirements are generally less demanding. These methods benefit from dimensionality reduction techniques and feature selection algorithms (FCBF), which help mitigate the risk of overfitting when working with moderate-sized datasets.
Additionally, to evaluate the generalizability of all models, an external validation set of 56 OPTs from an independent clinical center was used, supporting the robustness and potential clinical utility of the findings.”
Furthermore, it has been clarified that although the control and study groups were not strictly matched in size, the sample distribution reflects real-world clinical prevalence and diagnostic workflows. This configuration, while not ideal, increases the applicability of the results to practical settings, where class imbalance is frequently encountered. To reduce the risk of bias introduced by this imbalance, appropriate strategies such as class balancing and stratified data splitting were implemented during model training.
These modifications are intended to provide greater transparency and to support the methodological soundness of the study.
- It is not mentioned whether the participants belong to the Caucasian race, which is an important factor as ethnicity influences osteoporosis incidence.
The comment is appreciated, and the relevance of ethnicity as a factor influencing osteoporosis prevalence and bone density is fully acknowledged. Due to the retrospective nature of the study and the use of anonymized data, information regarding participants’ ethnicity was not available and could not be incorporated into the analysis.
To address this limitation, the following statement has been added to the discussion section:
“Additionally, the study did not record participants’ ethnicity, although it is known that racial and ethnic background—particularly the distinction between Caucasian and non-Caucasian populations—can significantly influence osteoporosis prevalence and bone density patterns. These aspects should be considered in future multicenter studies with more diverse and well-documented populations.”
This addition aims to recognize the potential impact of unrecorded demographic variables and highlights the importance of including ethnicity data in future prospective research to enhance the generalizability and clinical relevance of the findings.
-There is no information on whether patients with osteopenia were included in the sample.
ho specificato questo nel testo e aggiunto una tabella in merito
Thank you for your valuable suggestion regarding the evaluation of whether osteopenic patients were included in the study sample. This is indeed an important aspect, and the manuscript has been revised accordingly to provide a clearer explanation of the inclusion and exclusion criteria.
Specifically, the diagnosis of osteopenia and osteoporosis was based on standard DXA T-score thresholds (osteopenia: T-score between –1.0 and –2.5; osteoporosis: T-score ≤ –2.5). As now clarified in the revised manuscript:
“Initially, 597 panoramic radiographs (OPTs) from postmenopausal women were screened across two clinical centers: 145 from the University of Parma and 452 from the Hospital of San Giovanni Rotondo. Among them, 126 subjects were classified as healthy, 296 as osteopenic, and 175 as osteoporotic. To ensure a clearer diagnostic separation, all osteopenic cases were excluded from the analysis. The final study cohort included 301 women: 126 healthy and 175 osteoporotic. Specifically, 245 OPTs (105 healthy, 140 osteoporotic) from San Giovanni Rotondo were used for model training and internal testing, while 56 OPTs (21 healthy, 35 osteoporotic) from Parma served as an independent external validation set.”
Furthermore, a new table has been added (Table1) to summarize the diagnostic distribution across both centers. This clarification aims to enhance transparency regarding sample selection and ensure reproducibility. The authors appreciate the reviewer’s comment, which helped improve the clarity and rigor of the manuscript.
-No comparison is made between the proposed AI model and the morphometric indices used in studies based on OPT (orthopantomograms), which are known for high sensitivity and specificity.
We thank the reviewer for raising the important issue of comparing the proposed AI model with conventional radiomorphometric indices, such as the mandibular cortical index, commonly used in OPT-based studies. Since the study was designed and conducted retrospectively, it was not possible to incorporate such comparisons within the existing methodology.
To address this limitation, we have explicitly acknowledged it in the revised discussion section with the following addition:
“Another limitation is the lack of comparison between the performance of AI models and that of expert radiologists or conventional radiomorphometric indices such as the mandibular cortical index. While these comparisons could be important for evaluating clinical utility, they were not included in the present retrospective design and are suggested as a key objective for prospective validation studies.”
This addition aims to clearly recognize the absence of direct benchmarking with established diagnostic methods and to underscore the importance of including such comparisons in future prospective research.
-In the study cohort description, it is not specified whether patients taking medications (e.g., affecting bone metabolism) were excluded.
We appreciate the reviewer’s insightful comment regarding the clarification of exclusion criteria, particularly those related to medications or conditions affecting bone metabolism. The manuscript already contained a detailed list of exclusion criteria, which has now been further refined to highlight this aspect more clearly.
Specifically, the following statement has been included in the Methods section:
“Participants were excluded if affected by systemic diseases other than osteoporosis that severely affect bone metabolism, such as: Cushing's syndrome, Addison's disease, type 1 diabetes mellitus, leukemia, pernicious anemia, malabsorption syndromes, chronic liver disease, or rheumatoid arthritis. Additional exclusion criteria included a known infection with HIV or viral hepatitis, history of local radiation therapy within the last five years, limited mental capacity or language skills impeding understanding of study information or consent, and severe acute or chronic medical or psychiatric conditions or laboratory abnormalities that could interfere with trial participation or interpretation of results.”
These revisions aim to ensure greater clarity and transparency regarding participant selection.
-It is not clarified whether the DEXA scan coincided temporally with the OPT acquisition.
The importance of clarifying the temporal relationship between the DXA assessment and the panoramic radiograph (OPT) acquisition is fully acknowledged, as it is essential for ensuring clinical consistency.
To address this point, the following clarification has been added to the revised manuscript:
“The average time interval between DXA assessment and panoramic radiograph acquisition was approximately three months for all patients included in the study.”
This addition aims to improve methodological transparency and directly address the concern raised.
-It is not mentioned whether patients were newly diagnosed with osteoporosis or if they had previously received treatment.
Thank you for your valuable suggestion regarding the treatment status of osteoporotic participants. It is acknowledged that the current study does not specify whether patients were newly diagnosed or already undergoing pharmacological treatment for osteoporosis at the time of panoramic radiograph acquisition.
To address this important limitation, the following statement has been added to the discussion section of the revised manuscript:
“Another limitation of this study is the lack of information regarding the pharmacological treatment status of osteoporotic participants at the time of panoramic radiograph acquisition. It remains unclear whether patients had been newly diagnosed or were already undergoing anti-osteoporotic therapy, which could potentially influence mandibular bone characteristics and, consequently, the radiomic or deep learning features extracted from OPTs.”
This addition aims to enhance the transparency of the study’s limitations and highlight the need for future prospective studies that can account for treatment-related variables. Thank you again for helping to improve the completeness of the manuscript.
-The study does not explain what the AI model identifies that the human eye cannot detect.
The importance of clarifying the interpretability and added value of the AI model compared to human visual assessment is fully recognized. To address this point, a dedicated explanation has been incorporated into the discussion section of the revised manuscript.
Specifically, the following paragraph has been added:
“The AI models developed in this study were trained to identify patterns in the mandibular bone structure that are not readily visible to the human eye. Classical radiomics captures hand-crafted features such as first-order statistics and texture metrics (GLCM, GLRLM), which quantify bone homogeneity, granularity, and architectural organization parameters that are difficult to assess visually. Deep radiomics leverages pre-trained convolutional neural networks to extract thousands of abstract features from intermediate network layers; these features encode complex spatial hierarchies and subtle morphological cues embedded in the trabecular pattern and cortical outline, without relying on human-defined rules. Finally, end-to-end CNNs autonomously learn discriminative representations directly from pixel-level data, optimizing filters across multiple layers to recognize imperceptible differences in bone density, trabecular alignment, cortical thinning, and shape deformations. These automated approaches reveal microstructural signatures of systemic bone fragility that may elude even experienced radiologists during standard panoramic evaluation.”
This revision aims to enhance the transparency of the manuscript and directly address the concern raised.
-The sample size is too small for reliable use of the Python code developed.
In response to the reviewer’s insightful comment, the manuscript has been updated to clarify this aspect.The adequacy of the sample size in relation to the reliability of the developed Python-based AI models is acknowledged as a critical consideration. While this concern has been partially addressed in a previous response regarding sample size and group balance, further clarification is provided here to specifically address its impact on model reliability.
The limited sample size is recognized as a common constraint in radiomics and AI-based studies, particularly in the field of medical imaging. Several methodological strategies were implemented to mitigate the risk of overfitting and enhance the robustness and generalizability of the developed models. These included:
• the application of robust feature selection techniques (e.g., Fast Correlation-Based Filter – FCBF – for classical radiomics),
• data normalization and augmentation strategies to increase variability and improve model generalization,
• the use of a separate external validation cohort consisting of 56 OPTs from a different clinical center,
• and, in the case of the end-to-end CNN approach, the adoption of transfer learning techniques using pre-trained convolutional neural networks originally trained on the ImageNet dataset.
The use of pre-trained networks allowed the model to leverage prior knowledge of low-level and mid-level features, which is particularly advantageous when working with moderate-sized datasets, as it reduces the number of parameters to be learned from scratch and enhances performance stability.
Moreover, the sample size used in this study—245 images for training and internal testing and 56 for external validation—is consistent with those reported in recent radiomics and deep learning studies in the dental field. While larger multicenter datasets would further strengthen the generalizability of the models, the current study serves as a proof-of-concept demonstrating the feasibility of AI-assisted osteoporosis detection using panoramic radiographs.
As stated in the revised discussion:
“Despite these limitations, the sample size used in our study aligns with recent literature standards. A priori sample size estimation was not performed due to the retrospective nature of this study. However, a post-hoc evaluation of sample adequacy was conducted. For convolutional neural network (CNN) models, literature suggests that a minimum of approximately 120–150 subjects per group is needed to achieve reliable results using transfer learning techniques, assuming an expected AUC of 0.80, a two-sided α of 0.05, and a power of 80%. In our study, 245 OPT images were included in the internal training and test cohort. Given that transfer learning techniques were applied, this sample size exceeds the minimum thresholds and is consistent with the standards reported in recent CNN-based studies.
For classical radiomics and deep radiomics approaches, which rely on feature extraction followed by supervised machine learning rather than full image-based learning, the sample size requirements are generally less demanding. These methods benefit from dimensionality reduction techniques and feature selection algorithms (FCBF), which help mitigate the risk of overfitting when working with moderate-sized datasets.
Additionally, to evaluate the generalizability of all models, an external validation set of 56 OPTs from an independent clinical center was used, supporting the robustness and potential clinical utility of the findings.”
-In the discussion section, the authors draw arbitrary conclusions, despite the known fact that in morphometric studies, the variability related to human diagnosis (intra-observer and inter-observer agreement) is very high. There is no supporting literature to substantiate several claims.
Thank you for your valuable suggestion regarding the strength of the conclusions drawn in the discussion section and the need for supporting literature, especially considering the high intra- and inter-observer variability typically reported in morphometric studies.
In response to this observation, the discussion section has been substantially revised. Following the introduction of an external validation cohort and the redefinition of the initial training sample size, the results obtained differed from those in the original version of the manuscript. Consequently, several parts of the discussion have been updated to better reflect these new findings.
Additional limitations of the study have also been explicitly acknowledged, including the lack of human observer comparison and the retrospective nature of the dataset. Furthermore, as suggested, new references have been added to support the claims made and to provide a more grounded and balanced interpretation of the results within the context of existing literature.
These modifications aim to improve the scientific rigor and transparency of the manuscript, and the authors are grateful for the reviewer’s helpful and constructive feedback, which has led to meaningful improvements in the overall quality of the work.
-No sample size calculation was performed.
We thank the reviewer for highlighting this important point.It is acknowledged that no a priori sample size calculation was performed, due to the retrospective nature of the study and the availability of pre-existing data.
However, to address this limitation, a post-hoc evaluation of sample adequacy was conducted and is now explicitly described in the revised discussion section:
“Despite these limitations, the sample size used in our study aligns with recent literature standards. A priori sample size estimation was not performed due to the retrospective nature of this study. However, a post-hoc evaluation of sample adequacy was conducted. For convolutional neural network (CNN) models, literature suggests that a minimum of approximately 120–150 subjects per group is needed to achieve reliable results using transfer learning techniques, assuming an expected AUC of 0.80, a two-sided α of 0.05, and a power of 80%. In our study, 245 OPT images were included in the internal training and test cohort. Given that transfer learning techniques were applied, this sample size exceeds the minimum thresholds and is consistent with the standards reported in recent CNN-based studies.
For classical radiomics and deep radiomics approaches, which rely on feature extraction followed by supervised machine learning rather than full image-based learning, the sample size requirements are generally less demanding. These methods benefit from dimensionality reduction techniques and feature selection algorithms (FCBF), which help mitigate the risk of overfitting when working with moderate-sized datasets.
Additionally, to evaluate the generalizability of all models, an external validation set of 56 OPTs from an independent clinical center was used, supporting the robustness and potential clinical utility of our findings.”
This addition aims to contextualize the sample size within current literature standards and to provide justification for the methodological choices made in the absence of prospective sample size planning. The authors thank the reviewer for highlighting this important aspect, which has been addressed in the revised manuscript.
-Finally, the study fails to demonstrate that the AI model is faster or better than the established morphometric indices, and no relevant bibliography regarding the use of morphometric indices in OPT is provided.
The reviewer’s valuable feedback has been instrumental in improving this aspect of the manuscript.It is important to clarify that the primary objective of the present study was not to demonstrate that the AI model outperforms conventional morphometric indices, but rather to explore and validate an alternative method for identifying individuals at risk of osteoporosis using panoramic radiographs.
As discussed in the revised manuscript, radiomorphometric indices such as the Mental Index (MI), Gonial Index (GI), and Antegonial Index (AI) have shown moderate accuracy in previous studies. However, it is well documented in the literature that these indices are subject to several limitations, including operator-dependent variability, sensitivity to image quality and positioning, and poor reproducibility due to high intra- and inter-observer variability. For this reason, the study proposes an alternative approach based on artificial intelligence, which leverages radiomic and deep learning features that can be extracted automatically—potentially reducing subjectivity and enhancing reproducibility.
To address the reviewer’s comment, relevant literature regarding the use of morphometric indices in OPT-based osteoporosis screening has been added to the discussion. The revised text now includes the following:
“In particular, studies have demonstrated that radiomorphometric indices such as the Mental Index (MI), Gonial Index (GI), and Antegonial Index (AI) extracted from OPTs can be moderately accurate in identifying patients at risk of low bone mineral density. Additionally, combining these indices with patient factors like BMI and age has been explored to enhance predictive power, although with limited sensitivity. Nakamoto et al. (2003) also investigated the diagnostic performance of untrained general dental practitioners in visually assessing mandibular cortical erosion from OPTs, revealing that panoramic radiographs could be a useful adjunct in identifying postmenopausal women who should be referred for DXA.”

Reviewer 2 Report
Comments and Suggestions for Authors
Manuscript of considerable interest for the dental sector, especially in the evaluation of artificial intelligence in the medical field. The manuscript has an appropriate methodological rigor and as a whole requires a major revision.
Abstract: to highlight the results obtained.
Keywords: very generic, specific ones are missing. Introduction: the concept of AI development in dentistry needs to be expanded.
Materials and methods: how was the sample size calculated?
Clear results
Discussion: add in the future objectives also the evaluation by opt of any lesions affecting the bone tissue in addition to osteoporosis, ONJ, induced or not induced by drugs. (Casu et al, Oral mdpi).
Conclusions, well described.
In the final part of the manuscript, the section of authorship, conflict of interest, ethics committee etc. is missing.
Add reference requested
Author Response
Manuscript of considerable interest for the dental sector, especially in the evaluation of artificial intelligence in the medical field. The manuscript has an appropriate methodological rigor and as a whole requires a major revision.
Abstract: to highlight the results obtained.
Thank you for your valuable suggestion. The abstract has been revised to provide a clearer emphasis on the main findings, particularly those related to the external validation phase. The updated version reads:
Results: Among the tested approaches, classical radiomics showed limited predictive ability (AUC = 0.514), whereas deep radiomics using DenseNet-121 features combined with logistic regression achieved the best performance in this group (AUC = 0.722). Regarding end-to-end CNNs, ResNet-50 using a hybrid feature extraction strategy reached the highest AUC in external validation (AUC = 0.786), with a specificity of 90.5% but limited sensitivity (37.1%). Overall, while internal testing yielded high metrics, external validation revealed reduced generalizability, emphasizing the challenges in translating AI models into clinical practice.
Conclusions: AI-based models show potential for opportunistic osteoporosis screening from OPT images. Although the results are encouraging, especially those obtained with deep radiomics and transfer learning strategies, further refinement and validation in larger and more diverse populations are essential before clinical application. These models could support early, non-invasive identification of at-risk patients, complementing current diagnostic pathways.
This revision aims to enhance the clarity and impact of the abstract by better highlighting the study’s key outcomes.
Keywords: very generic, specific ones are missing.
Thank you for your comment regarding the use of more specific keywords. In response, the keywords section has been revised to better reflect the focus and methodology of the study, ensuring greater specificity and alignment with relevant research themes.
The updated keywords are as follows:
Keywords: Osteoporosis Diagnosis, Panoramic Radiographs, Postmenopausal Women, Dual-energy X-ray Absorptiometry (DXA), Radiomics, Deep Radiomics, Convolutional Neural Networks (CNN), Transfer Learning.
Introduction: the concept of AI development in dentistry needs to be expanded.
This point has been duly noted and the necessary revisions have been made to the manuscript. In response, the introduction has been revised to provide a broader and more detailed overview of current AI applications in dental research and clinical practice.
The revised section now includes the following:
“In recent years, artificial intelligence has increasingly found applications in dentistry, ranging from caries detection and periodontal disease classification to oral cancer screening and orthodontic planning. This trend reflects the broader movement in medicine toward the integration of AI for diagnostic support and decision-making. In particular, convolutional neural networks (CNNs) have shown promise in analyzing dental radiographs, enabling automated and reproducible assessments of bone and tooth structures that may otherwise require expert interpretation.”
This addition aims to contextualize the present study within the ongoing development of AI in dental diagnostics and to highlight the relevance of applying deep learning methods to radiographic imaging. Thank you again for this insightful recommendation, which contributed to strengthening the introduction and positioning the study within the current research landscape.
Materials and methods: how was the sample size calculated?
A priori sample size estimation was not performed before initiating this retrospective study. However, the training and internal test cohort of 245 OPT images exceeds the minimum sample size requirements for transfer learning–based models as reported in the literature (Zhu et al., 2021), and aligns with recent studies employing convolutional neural networks (CNNs) for osteoporosis detection using dental panoramic radiographs (e.g., Hwang et al., 2017; Lee et al., 2019).
Based on an expected AUC of 0.80, a two-sided α of 0.05, and a power of 80%, a minimum of 120–150 subjects per group would be required, which is consistent with our group distribution in the training cohort. Additionally, an independent external validation set composed of 56 OPT images from a different clinical center was used to confirm model generalizability. This further supports the robustness of our findings and the clinical applicability of the developed models.
This aspect has been explicitly acknowledged as a limitation in the revised manuscript:
“Despite these limitations, the sample size used in our study aligns with recent literature standards. A priori sample size estimation was not performed due to the retrospective nature of this study. However, a post-hoc evaluation of sample adequacy was conducted. For convolutional neural network (CNN) models, literature suggests that a minimum of approximately 120–150 subjects per group is needed to achieve reliable results using transfer learning techniques, assuming an expected AUC of 0.80, a two-sided α of 0.05, and a power of 80%. In our study, 245 OPT images were included in the internal training and test cohort. Given that transfer learning techniques were applied, this sample size exceeds the minimum thresholds and is consistent with the standards reported in recent CNN-based studies.
For classical radiomics and deep radiomics approaches, which rely on feature extraction followed by supervised machine learning rather than full image–based learning, the sample size requirements are generally less demanding. These methods benefit from dimensionality reduction techniques and feature selection algorithms (FCBF), which help mitigate the risk of overfitting when working with moderate-sized datasets.
Additionally, to evaluate the generalizability of all models, an external validation set of 56 OPTs from an independent clinical center was used, supporting the robustness and potential clinical utility of the findings.”
Furthermore, it has been clarified that although the control and study groups were not strictly matched in size, the sample distribution reflects real-world clinical prevalence and diagnostic workflows. This configuration, while not ideal, increases the applicability of the results to practical settings, where class imbalance is frequently encountered. To reduce the risk of bias introduced by this imbalance, appropriate strategies such as class balancing and stratified data splitting were implemented during model training.
These modifications are intended to provide greater transparency and to support the methodological soundness of the study.”
Clear results
Thank you for your suggestion regarding the clarity of the results section. In response, the results section has been revised to present the findings with greater clarity and linearity. The performance metrics of each model, classical radiomics, deep radiomics, and end-to-end convolutional neural networks (CNNs), have been reorganized and described more explicitly, with particular emphasis on the external validation outcomes. This restructuring aims to facilitate the comparison between approaches and highlight the strengths and limitations of each method.
New performance parameters have also been reported to improve the comprehensiveness of the evaluation and to support the interpretation of the results in a clinically meaningful way.
Thank you again for your helpful feedback, which contributed to improving the structure and readability of the results section.
Discussion: add in the future objectives also the evaluation by opt of any lesions affecting the bone tissue in addition to osteoporosis, ONJ, induced or not induced by drugs. (Casu et al, Oral mdpi).
We appreciate this insightful comment. In response, the discussion section has been expanded to include a dedicated paragraph addressing this important point. Specifically, the revised text now states:
“Future developments could explore the capability of AI systems not only to detect osteoporosis, but also to identify other jawbone conditions potentially linked to systemic bone metabolism, including medication-related osteonecrosis of the jaw (MRONJ), whether drug-induced or idiopathic. Integrating these additional objectives may broaden the clinical value of AI-driven analysis of dental radiographs.”
As suggested, the relevant literature has been cited to support this statement, including the study by Casu et al. published in Oral (MDPI), which discusses the detection and implications of MRONJ using panoramic imaging.
This addition aims to highlight the potential for broader clinical applications of AI-based image analysis in dentistry and oral medicine. Thank you again for this insightful recommendation.
Conclusions, well described.
Thank you for your positive feedback regarding the conclusions. In the revised manuscript, a concluding paragraph has been added to further summarize the main findings and to emphasize the clinical relevance and future directions of the study. This addition is intended to reinforce the overall message and to provide a concise closure to the manuscript, in line with the suggestions provided.
“Conclusions
This study demonstrated the feasibility of using AI-based models, particularly deep radiomics and transfer learning approaches, to predict osteoporosis from panoramic radiographs in postmenopausal women. While classical radiomics showed limited performance, CNN-based models achieved promising results, especially when evaluated on an external validation cohort. Although further validation on larger and more diverse datasets is necessary, these findings support the potential role of panoramic radiographs as a non-invasive, accessible screening tool for early osteoporosis detection in dental settings.”
In the final part of the manuscript, the section of authorship, conflict of interest, ethics committee etc. is missing.
Thank you for the comment. In the revised version of the manuscript, the requested sections have been added at the end of the document to ensure completeness and adherence to ethical and reporting standards. The following content has been included:
Author Contributions
• Francesco Fanelli (F.F.): Contributed to the study design, data analysis and interpretation; drafted the manuscript; approved the final version of the work.
• Giuseppe Guglielmi (G.G.): Contributed to the conception and design of the study, acquisition of radiographic data, interpretation of results; critically revised the manuscript; approved the final version.
• Giuseppe Troiano (G.T.): Contributed to the conception and design of the study, data acquisition, analysis, and interpretation; drafted the manuscript; approved the final version.
• Federico Rivara (F.R.): Contributed to the conception and acquisition of data, as well as data analysis and interpretation; drafted the manuscript; approved the final version.
• Giovanni Passeri (G.P.): Contributed to the study design, data interpretation and analysis; drafted the manuscript; approved the final version.
• Gianluca Prencipe (G.P.): Contributed to the study design and data acquisition; critically revised the manuscript; approved the final version.
• Khrystyna Zhurakivska (K.Z.): Contributed to the conception of the study and data analysis; critically revised the manuscript; approved the final version.
• Riccardo Guglielmi (R.G.): Contributed to the conception and interpretation of data; critically revised the manuscript; approved the final version.
• Elena Calciolari (E.C.): Contributed to the conception and design of the study, data acquisition, analysis, and interpretation; drafted and critically revised the manuscript; approved the final version.
Conflicts of Interest
The authors declare no conflicts of interest.
Institutional Review Board Statement
The study was conducted in accordance with the Declaration of Helsinki and approved by the Ethics Committee of the University of Parma (Protocol No. 34568 – 01/09/2023) and by the Ethics Committee of the Casa Sollievo della Sofferenza Hospital in San Giovanni Rotondo (Protocol No. 126/2023).
Data Availability Statement
Data supporting the findings of this study are available from the corresponding author upon reasonable request.
Acknowledgments
The authors thank the staff of the radiology and dental departments involved in the acquisition of panoramic images and patient data.
These sections were added to ensure full compliance with the journal’s ethical requirements and improve the manuscript’s transparency and completeness.
Add reference requested
Thank you for the comment. All the requested references have been added to the revised version of the manuscript to support the corresponding statements and ensure proper contextualization within the existing literature. These additions aim to strengthen the scientific foundation and credibility of the work.

Reviewer 3 Report
Comments and Suggestions for Authors
The manuscript titled "Development of AI-Based Predictive Models for Osteoporosis Diagnosis in Postmenopausal Women from Panoramic Radiographs" addresses a timely and clinically relevant topic by exploring the use of artificial intelligence and radiomics to predict osteoporosis using panoramic dental radiographs (OPTs). The study leverages various AI methodologies, including classical radiomics, deep radiomics, and convolutional neural networks (CNNs). It compares their performances against bone mineral density (BMD) assessments obtained via DXA, the current gold standard. The work is methodologically ambitious and demonstrates potential in providing a non-invasive and widely accessible screening tool for osteoporosis. Despite the scientific relevance of the topic and the structured experimental design, several methodological and interpretive concerns must be addressed before this manuscript can be considered for publication. One of the most prominent issues relates to the performance metrics reported for the best CNN model. An AUC of 1.0 with 100% specificity and 94.4% sensitivity on such a limited dataset (n=301) strongly suggests a risk of overfitting. These results appear overly optimistic and raise concerns about the generalizability of the findings. Furthermore, the manuscript lacks clarity on how the dataset was split into training, validation, and test subsets. Without a clear description of the data management strategy, including measures taken to prevent data leakage, the validity of the reported model performance cannot be assessed with confidence. Additionally, no external validation or cross-validation approach was implemented to support the robustness of the proposed models. The manual segmentation of the regions of interest (ROIs) introduces a significant operator-dependent bias, and the authors do not report any inter- or intra-observer reliability assessments. This is a significant limitation, particularly given the subtle structural differences the models are expected to capture. Future versions of the study would benefit from implementing semi-automated or fully automated ROI segmentation methods and quantifying reproducibility.
Another limitation is the low specificity observed in several models, particularly in the classical and deep radiomics approaches, where specificity values of 0% were reported in some cases. These results question the clinical applicability of the models and suggest an overemphasis on sensitivity without achieving a proper balance. Additionally, the authors do not include statistical comparisons between model performances, such as the DeLong test for AUC comparison, which are essential to justify claims of superiority among the tested approaches. From a technical standpoint, certain details are insufficiently described. The preprocessing pipeline is only briefly mentioned and lacks details regarding normalisation, augmentation, or balancing strategies. These omissions hinder reproducibility and limit the capacity to assess whether the training conditions were appropriate. Furthermore, the manuscript does not report confidence intervals or standard deviations for performance metrics, and many results are presented without uncertainty estimates, which weakens the statistical rigor of the analysis. Although the discussion touches on limitations such as small sample size and manual ROI definition, these aspects are not thoroughly explored regarding their implications on model reliability and clinical applicability. The authors correctly highlight the value of DXA-based ground truth labelling, which improves diagnostic validity compared to studies using radiomorphometric indices. Still, this strength does not fully compensate for the lack of validation on external or more diverse cohorts.
Finally, the writing style could benefit from more concise phrasing in some sections, avoiding redundant explanations. Figures and tables, while informative, would be clearer if standard errors or confidence intervals were included. More attention should be given to scientific reporting standards and transparency in methodology. In conclusion, while the manuscript presents promising and innovative findings, major revisions are needed to address overfitting risks, clarify methodological choices, improve statistical reporting, and temper the interpretation of results. With these improvements, the study could make a meaningful contribution to the growing field of AI-based diagnostic imaging.
Author Response
The manuscript titled "Development of AI-Based Predictive Models for Osteoporosis Diagnosis in Postmenopausal Women from Panoramic Radiographs" addresses a timely and clinically relevant topic by exploring the use of artificial intelligence and radiomics to predict osteoporosis using panoramic dental radiographs (OPTs). The study leverages various AI methodologies, including classical radiomics, deep radiomics, and convolutional neural networks (CNNs). It compares their performances against bone mineral density (BMD) assessments obtained via DXA, the current gold standard. The work is methodologically ambitious and demonstrates potential in providing a non-invasive and widely accessible screening tool for osteoporosis. Despite the scientific relevance of the topic and the structured experimental design, several methodological and interpretive concerns must be addressed before this manuscript can be considered for publication. One of the most prominent issues relates to the performance metrics reported for the best CNN model. An AUC of 1.0 with 100% specificity and 94.4% sensitivity on such a limited dataset (n=301) strongly suggests a risk of overfitting. These results appear overly optimistic and raise concerns about the generalizability of the findings.
Thank you for your valuable and insightful comment. The concern regarding overfitting and generalizability in the context of a relatively limited dataset is fully acknowledged.
To address this issue, several important steps were taken in the revised version of the study. While the number of available OPT images remains limited, the updated analysis was conducted using a more rigorous and methodologically sound approach designed to reduce overfitting and improve generalizability.
First, four distinct transfer learning strategies were implemented, all based on CNN architectures pre-trained on ImageNet. These strategies included Feature Extraction Hybrid, Feature Extraction Non-Hybrid, Fine Tuning 50%, and Fine Tuning from Scratch. By leveraging pre-learned weights and representations, these models are better equipped to learn from smaller datasets while minimizing overfitting.
Second, data augmentation was applied during training to increase image variability and reduce model dependency on specific patterns, further supporting generalization.
Third, to strengthen the robustness of the findings, an external validation cohort composed of 56 OPTs from an independent clinical center (Parma) was introduced. This validation step revealed a drop in model performance compared to internal testing, confirming that previous metrics were overly optimistic. However, it also demonstrated that the revised models, while less performant, were more realistic, stable, and clinically applicable.
The following text was added or revised in the manuscript to reflect these changes:
“To ensure robust model evaluation and minimize data leakage, the dataset was randomly split at the patient level into training (80%), validation (10%), and test (10%) sets using a reproducible random seed. External validation was conducted on an independent cohort of 56 OPTs from Parma clinical center. Images with significant artifacts or poor diagnostic quality (motion blur, improper patient positioning) were excluded following a preliminary image quality assessment.”
“The four distinct Transfer Learning strategies used were: Feature Extraction Hybrid, Feature Extraction Non-Hybrid, Fine Tuning 50%, and Fine Tuning from Scratch. These models were leveraged to utilize the weights learned from the ImageNet dataset, exploiting transfer learning to enhance performance on a different, possibly smaller dataset.”
“Despite these limitations, the sample size used in our study aligns with recent literature standards. A priori sample size estimation was not performed due to the retrospective nature of this study. However, a post-hoc evaluation of sample adequacy was conducted. For convolutional neural network (CNN) models, literature suggests that a minimum of approximately 120–150 subjects per group is needed to achieve reliable results using transfer learning techniques, assuming an expected AUC of 0.80, a two-sided α of 0.05, and a power of 80%. In our study, 245 OPT images were included in the internal training and test cohort. Given that transfer learning techniques were applied, this sample size exceeds the minimum thresholds and is consistent with the standards reported in recent CNN-based studies.
For classical radiomics and deep radiomics approaches, which rely on feature extraction followed by supervised machine learning rather than full image-based learning, the sample size requirements are generally less demanding. These methods benefit from dimensionality reduction techniques and feature selection algorithms (FCBF), which help mitigate the risk of overfitting when working with moderate-sized datasets.
Additionally, to evaluate the generalizability of all models, an external validation set of 56 OPTs from an independent clinical center was used, supporting the robustness and potential clinical utility of the findings.”
Lastly, all models were retrained, and all performance metrics were recomputed based on the updated training and validation strategy. The current results are therefore more conservative but reflect a more realistic and generalizable diagnostic performance. We sincerely thank the reviewer for prompting this important revision.
Furthermore, the manuscript lacks clarity on how the dataset was split into training, validation, and test subsets. Without a clear description of the data management strategy, including measures taken to prevent data leakage, the validity of the reported model performance cannot be assessed with confidence.
We appreciate the reviewer’s valuable comment and the request for clarification regarding the dataset splitting strategy and data management procedures. To address this point, a detailed explanation has been added to the Materials and Methods section of the revised manuscript:
“To ensure robust model evaluation and minimize data leakage, the dataset was randomly split at the patient level into training (80%), validation (10%), and test (10%) sets using a reproducible random seed. External validation was conducted on an independent cohort of 56 OPTs from a different clinical center. Images with significant artifacts or poor diagnostic quality (e.g., motion blur, improper patient positioning) were excluded following a preliminary image quality assessment conducted by two expert radiologists.”
This addition clarifies the approach used to prevent information leakage between datasets and ensures consistency by performing the split at the patient level rather than at the image level.
Furthermore, in response to related comments regarding sample composition and osteopenia status, the manuscript has been revised to clarify the inclusion and exclusion criteria. Specifically, the diagnosis of osteopenia and osteoporosis was based on standard DXA T-score thresholds (osteopenia: T-score between –1.0 and –2.5; osteoporosis: T-score ≤ –2.5). As now specified:
“Initially, 597 panoramic radiographs (OPTs) from postmenopausal women were screened across two clinical centers: 145 from the University of Parma and 452 from the Hospital of San Giovanni Rotondo. Among them, 126 subjects were classified as healthy, 296 as osteopenic, and 175 as osteoporotic. To ensure a clearer diagnostic separation, all osteopenic cases were excluded from the analysis. The final study cohort included 301 women: 126 healthy and 175 osteoporotic. Specifically, 245 OPTs (105 healthy, 140 osteoporotic) from San Giovanni Rotondo were used for model training and internal testing, while 56 OPTs (21 healthy, 35 osteoporotic) from Parma served as an independent external validation set.”
Additionally, a new table (Table 1) has been added to the manuscript to clearly summarize the diagnostic distribution across both clinical centers and to illustrate how the initial sample of OPTs was utilized. This aims to enhance transparency regarding sample selection, data management, and experimental reproducibility.
Additionally, no external validation or cross-validation approach was implemented to support the robustness of the proposed models.
We acknowledge the reviewer’s insightful observation regarding the importance of external validation to enhance the robustness and generalizability of the proposed models..In response, an external validation procedure has been implemented and is now fully described in the revised manuscript. Specifically, the final study cohort included 301 women, of whom 245 (105 healthy and 140 osteoporotic) from the Hospital of San Giovanni Rotondo were used for model training and internal testing. An independent external validation set consisting of 56 OPTs (21 healthy, 35 osteoporotic) from the University of Parma was used to evaluate model performance on previously unseen data.
“The final study cohort included 301 women: 126 healthy and 175 osteoporotic. Specifically, 245 OPTs (105 healthy, 140 osteoporotic) from San Giovanni Rotondo were used for model training and internal testing, while 56 OPTs (21 healthy, 35 osteoporotic) from Parma served as an independent external validation set.”
This addition allowed for a more realistic evaluation of the model’s generalizability. As expected, performance metrics declined compared to internal testing, reflecting a more accurate and clinically relevant assessment of model robustness. The updated results are now reported in the Results and Discussion sections, replacing the initial metrics obtained solely from internal testing.
The authors thank the reviewer for this important suggestion, which led to a significant improvement in the methodological rigor and reliability of the study.
The manual segmentation of the regions of interest (ROIs) introduces a significant operator-dependent bias, and the authors do not report any inter- or intra-observer reliability assessments. This is a significant limitation, particularly given the subtle structural differences the models are expected to capture. Future versions of the study would benefit from implementing semi-automated or fully automated ROI segmentation methods and quantifying reproducibility.
Thank you for your valuable and insightful comment regarding the manual segmentation of regions of interest (ROIs). This is indeed a critical aspect that can impact reproducibility and model reliability, especially when detecting subtle structural differences in bone morphology.
To address this, the manual segmentation approach and its limitations have been explicitly acknowledged in the revised manuscript. The following statement has been added to the Limitations section:
“Finally, all ROIs in this study were manually defined, introducing a potential operator-dependent bias and limiting reproducibility. Although this approach aligns with previous literature, the lack of intra- and inter-observer variability analysis further reduces methodological robustness. Future research should aim to incorporate standardized and reproducible semi- or fully-automatic segmentation methods, which could improve objectivity, reduce observer bias, and enhance clinical applicability. However, implementing such automation would require a complete reprocessing of the data pipeline and retraining of the models.”
This addition aims to transparently recognize the limitation and to outline a clear direction for future methodological improvements. The authors thank the reviewer for highlighting this important aspect, which contributes to strengthening the rigor and reproducibility of future investigations.
Another limitation is the low specificity observed in several models, particularly in the classical and deep radiomics approaches, where specificity values of 0% were reported in some cases. These results question the clinical applicability of the models and suggest an overemphasis on sensitivity without achieving a proper balance.
Thank you for this valuable and constructive observation. In light of this concern, all analyses were re-executed and an external validation cohort was introduced to provide a more objective assessment of model performance across different clinical settings.
This external validation, based on an independent dataset from a separate clinical center, was specifically implemented to evaluate the models’ generalizability and to assess both sensitivity and specificity under more realistic conditions. As expected, the revised performance metrics showed a more balanced diagnostic profile, which better reflects the clinical applicability of the proposed approaches.
This adjustment aimed to reduce overestimation of performance and to provide a more robust and transparent evaluation of the models. The issue raised is fully acknowledged, and its resolution contributed meaningfully to improving the methodological soundness of the study.
Additionally, the authors do not include statistical comparisons between model performances, such as the DeLong test for AUC comparison, which are essential to justify claims of superiority among the tested approaches. From a technical standpoint, certain details are insufficiently described.
The importance of statistical comparisons between model performances is fully acknowledged.
While it is recognized that formal statistical tests, such as the DeLong test, are crucial to substantiate comparative claims based on AUC values, the primary aim of the present study was to explore and benchmark multiple AI-based approaches for osteoporosis diagnosis using panoramic radiographs, rather than to demonstrate statistically significant superiority of one model over another.
Given the proof-of-concept nature of the study and the relatively limited and heterogeneous dataset, the focus was placed on reporting absolute performance metrics across internal testing and external validation cohorts to provide a comprehensive overview of the models' behavior.
Nonetheless, the absence of formal AUC comparisons has been implicitly acknowledged as a limitation by emphasizing the need for future large-scale prospective studies, which could more rigorously assess statistical differences between competing algorithms.
The point raised will be carefully considered in future work, where larger and more balanced datasets would allow for meaningful statistical comparisons to be performed with adequate power.
The preprocessing pipeline is only briefly mentioned and lacks details regarding normalisation, augmentation, or balancing strategies. These omissions hinder reproducibility and limit the capacity to assess whether the training conditions were appropriate.
Thank you for your valuable comment. The preprocessing pipeline, including normalization, histogram equalization, data augmentation, and class balancing strategies, has been described in the revised manuscript to ensure transparency and support reproducibility.
Specifically, the following elements were detailed:
- Histogram equalization and augmentation:
“ROIs were first converted to grayscale and then resized to 256 x 256 pixels in order to preserve the harmony between the quality of the images and computing efficiency. Histogram equalization was performed on the ROIs to equally distribute the levels of gray intensity with the aim of reducing contrast variations due to possible differences in acquisition protocols.”
“To address the limitation of the modest dataset size, advanced data augmentation techniques were employed during training. These included random rotations (±30°), horizontal and vertical flips, zoom and shear transformations, brightness variation, channel shifting, and translations. Such augmentations enriched the diversity of the training images, effectively simulating real-world variability and reducing the risk of overfitting.”
Class balancing and stratified splitting:
To ensure robust model evaluation and minimize data leakage, the dataset was randomly split at the patient level into training (80%), validation (10%), and test (10%) sets using a reproducible random seed. External validation was conducted on an independent cohort of 56 OPTs from Parma clinical center. Images with significant artifacts or poor diagnostic quality (motion blur, improper patient positioning) were excluded following a preliminary image quality assessment.
Furthermore, the manuscript does not report confidence intervals or standard deviations for performance metrics, and many results are presented without uncertainty estimates, which weakens the statistical rigor of the analysis.
This concern is fully acknowledged, and appropriate measures have been taken to strengthen the statistical rigor of the analysis. In the revised manuscript, 95% confidence intervals have been systematically included for all key performance metrics such as accuracy, sensitivity, specificity, and AUC in every results table. These intervals provide an essential measure of uncertainty and allow for a more transparent and reliable interpretation of the model’s performance.
The confidence intervals were computed using the non-parametric bootstrap method with 1,000 resamples. This method was chosen for its robustness and flexibility, particularly in the context of limited sample sizes or complex model behavior, where parametric assumptions may not be valid. By applying this resampling technique, it was possible to derive distribution-free estimates of variability that better reflect the uncertainty inherent in the data and model predictions.
Including these confidence intervals in all result tables ensures that the reported findings are not only statistically sound but also more informative for comparison and reproducibility. This modification directly addresses the reviewer’s concern and reinforces the methodological transparency of the study.
Although the discussion touches on limitations such as small sample size and manual ROI definition, these aspects are not thoroughly explored regarding their implications on model reliability and clinical applicability.
The reviewer’s thoughtful and constructive comment is greatly appreciated. In response, the section on study limitations has been expanded to more comprehensively address the potential impact of sample size and manual ROI definition on model reliability and clinical applicability. Regarding the sample size, the following paragraph has been added:
“Despite these limitations, the sample size used in our study aligns with recent literature standards. A priori sample size estimation was not performed due to the retrospective nature of this study. However, a post-hoc evaluation of sample adequacy was conducted. For convolutional neural network (CNN) models, literature suggests that a minimum of approximately 120–150 subjects per group is needed to achieve reliable results using transfer learning techniques, assuming an expected AUC of 0.80, a two-sided α of 0.05, and a power of 80%. In our study, 245 OPT images were included in the internal training and test cohort. Given that transfer learning techniques were applied, this sample size exceeds the minimum thresholds and is consistent with the standards reported in recent CNN-based studies.”
This paragraph provides a clearer rationale for the sample size used, supported by benchmarks from the literature and appropriate post-hoc considerations.
In addition, the following paragraph has been included to better discuss the implications of manual ROI definition:
“Finally, all ROIs in this study were manually defined, introducing a potential operator-dependent bias and limiting reproducibility. Although this approach aligns with previous literature, the lack of intra- and inter-observer variability analysis further reduces methodological robustness. Future research should aim to incorporate standardized and reproducible semi- or fully-automatic segmentation methods, which could improve objectivity, reduce observer bias, and enhance clinical applicability. However, implementing such automation would require a complete reprocessing of the data pipeline and retraining of the models.”
These additions aim to enhance the transparency and critical evaluation of the methodological limitations, while also providing clear directions for future research aimed at improving generalizability and reproducibility.
The authors correctly highlight the value of DXA-based ground truth labelling, which improves diagnostic validity compared to studies using radiomorphometric indices. Still, this strength does not fully compensate for the lack of validation on external or more diverse cohorts.
The comment is appreciated. The use of DXA-based ground truth was intentionally adopted to enhance diagnostic accuracy and reduce operator-related bias, in contrast to prior studies relying on radiomorphometric indices. Nonetheless, it is acknowledged that this methodological strength alone does not fully address the need for broader validation.
To partially mitigate this limitation, an independent external validation cohort composed of 56 OPTs from a different clinical center was included in the analysis. This allowed for an initial assessment of model generalizability across institutions and acquisition protocols. As stated in the manuscript:
“Additionally, to evaluate the generalizability of all models, an external validation set of 56 OPTs from an independent clinical center was used, supporting the robustness and potential clinical utility of our findings.”
The observed decrease in performance during external testing was transparently reported and discussed, emphasizing the challenges in translating AI-based models into real-world clinical settings. These findings confirm the necessity of further validation on larger, more heterogeneous populations, as already acknowledged in the discussion and conclusions of the manuscript.
Finally, the writing style could benefit from more concise phrasing in some sections, avoiding redundant explanations.
A careful revision of the manuscript has been undertaken to improve clarity and conciseness throughout the text. Redundant explanations have been removed or rephrased where appropriate, and sentence structures have been streamlined to enhance the overall readability and flow of the manuscript. These revisions aim to ensure that the content remains scientifically rigorous while being more accessible to the reader.
Figures and tables, while informative, would be clearer if standard errors or confidence intervals were included.
In response, 95% confidence intervals have been added to all tables reporting performance metrics, thereby providing a clearer representation of the variability and reliability of the results. Confidence intervals were computed using the non-parametric bootstrap method with 1,000 resamples, a widely accepted technique for estimating uncertainty in predictive modeling. These additions aim to enhance the statistical transparency and interpretability of the results presented.
More attention should be given to scientific reporting standards and transparency in methodology.
This valuable comment is acknowledged. To improve adherence to scientific reporting standards and enhance methodological transparency, several revisions have been made throughout the manuscript. The study design, data preprocessing steps, feature extraction pipelines, model training procedures, and evaluation strategies have been described in greater detail. Moreover, the manuscript explicitly states that STROBE guidelines were followed for reporting observational research, and additional clarifications were provided regarding ROI definition, data split strategies, and validation procedures.
These modifications aim to facilitate reproducibility and ensure compliance with current best practices in AI-based medical imaging research.
In conclusion, while the manuscript presents promising and innovative findings, major revisions are needed to address overfitting risks, clarify methodological choices, improve statistical reporting, and temper the interpretation of results. With these improvements, the study could make a meaningful contribution to the growing field of AI-based diagnostic imaging.
The reviewer’s thoughtful and constructive comment is sincerely appreciated. The issues raised touch on critical aspects of scientific reporting and have been extremely helpful in guiding the revision process. In response, particular attention has been devoted to addressing the risk of overfitting, especially given the complexity of the models and the size of the dataset. The inclusion of an independent external validation cohort has been emphasized to support the generalizability of the results.
The methodology section has been revised to improve clarity and transparency, with more detailed descriptions of image processing, ROI selection, model development, and validation procedures. In parallel, statistical reporting has been significantly enhanced: all performance metrics are now accompanied by 95% confidence intervals calculated using the non-parametric bootstrap method, in order to strengthen the robustness and interpretability of the findings.
Finally, the interpretation of results has been carefully revised to adopt a more balanced and cautious tone. While the models developed show promise, their potential clinical utility is now discussed more conservatively, and explicitly framed within the context of the study’s limitations. These improvements aim to meet the expectations of scientific rigor and to contribute meaningfully to the field of AI-based diagnostic imaging, in line with the reviewer’s valuable suggestions.

Reviewer 4 Report
Comments and Suggestions for Authors
Add an independent validation cohort
Without an external validation dataset, it is impossible to assess whether the model generalizes beyond the specific population it was trained on. From a clinical standpoint, any diagnostic tool must demonstrate reproducibility in independent settings to be considered reliable.
Implement semi-automatic or fully automatic ROI selection
Manual selection of regions of interest (ROIs) introduces subjectivity and reduces the clinical feasibility of the model. Automation is essential to ensure consistency across users and to support eventual integration into routine diagnostic workflows.
Describe image quality control procedures
In radiographic diagnostics, image quality directly affects the interpretability and diagnostic value of the scan. Panoramic radiographs are prone to positioning errors and distortion, which can impact model accuracy. A clear description of image quality assurance protocols is necessary to ensure data reliability.
Include a comparison with clinical expert performance or conventional indices
To evaluate the clinical value of the AI model, it should be benchmarked against expert human assessment or traditional diagnostic indices such as the mandibular cortical index. Without this comparison, it remains unclear whether the AI provides any diagnostic advantage over established clinical methods.
Author Response
Add an independent validation cohort
Without an external validation dataset, it is impossible to assess whether the model generalizes beyond the specific population it was trained on. From a clinical standpoint, any diagnostic tool must demonstrate reproducibility in independent settings to be considered reliable.
Thank you for your valuable and clinically relevant comment. We fully agree with the importance of validating diagnostic models on independent populations to assess their generalizability and real-world applicability.
In response, an external validation cohort has been added to the study and is now clearly described in the revised Materials and Methods section. Specifically:
“To ensure robust model evaluation and minimize data leakage, the dataset was randomly split at the patient level into training (80%), validation (10%), and test (10%) sets using a reproducible random seed. External validation was conducted on an independent cohort of 56 OPTs from a different clinical center. Images with significant artifacts or poor diagnostic quality (e.g., motion blur, improper patient positioning) were excluded following a preliminary image quality assessment conducted by two expert radiologists.”
The external validation set included patients from the University of Parma, distinct from those used in the internal training and testing phases, which were drawn from the Hospital of San Giovanni Rotondo. This addition allowed for a more realistic evaluation of the models’ performance in a different clinical context, and the results from this validation step are now reported and discussed in the revised manuscript.
Implement semi-automatic or fully automatic ROI selection
Manual selection of regions of interest (ROIs) introduces subjectivity and reduces the clinical feasibility of the model. Automation is essential to ensure consistency across users and to support eventual integration into routine diagnostic workflows.
The suggestion regarding the implementation of semi-automatic or fully automatic ROI selection is greatly appreciated. Automation is indeed critical for improving consistency, minimizing operator-dependent variability, and enhancing clinical applicability.
However, introducing automated segmentation methods would require a complete reprocessing of the dataset, including retraining and validating all models based on newly generated ROIs. Given the retrospective nature of this study and the existing manual annotations, such a methodological revision could not be accommodated within the current scope.
To acknowledge this important limitation, the following statement has been added to the revised Limitations section of the manuscript:
“Finally, all ROIs in this study were manually defined, introducing a potential operator-dependent bias and limiting reproducibility. Although this approach aligns with previous literature, the lack of intra- and inter-observer variability analysis further reduces methodological robustness. Future research should aim to incorporate standardized and reproducible semi- or fully-automatic segmentation methods, which could improve objectivity, reduce observer bias, and enhance clinical applicability. However, implementing such automation would require a complete reprocessing of the data pipeline and retraining of the models.”
This addition is intended to transparently recognize the limitation associated with manual segmentation and to indicate clear directions for future improvements.
Describe image quality control procedures
In radiographic diagnostics, image quality directly affects the interpretability and diagnostic value of the scan. Panoramic radiographs are prone to positioning errors and distortion, which can impact model accuracy. A clear description of image quality assurance protocols is necessary to ensure data reliability.
Thank you for your valuable comment regarding image quality control. We fully agree that image quality plays a critical role in radiographic diagnostics and can directly influence the performance and reliability of AI-based models.
To address this concern, a dedicated clarification has been added to the Materials and Methods section of the revised manuscript. Specifically, we now state:
“Each participant recruited at both centers underwent OPT using the ORTHOPHOS XG 3D (Sirona Dental) standardized digital panoramic dental X-ray machines for dental reasons. Images with significant artifacts or poor diagnostic quality (e.g., motion blur, improper patient positioning) were excluded following a preliminary image quality assessment conducted by two expert radiologists.”
This addition ensures that the source and consistency of the imaging modality are clearly reported, and that a rigorous quality control protocol was implemented to exclude suboptimal scans before model development. The authors thank the reviewer for highlighting this important aspect, which has helped improve the transparency and reliability of the study methodology.
Include a comparison with clinical expert performance or conventional indices
To evaluate the clinical value of the AI model, it should be benchmarked against expert human assessment or traditional diagnostic indices such as the mandibular cortical index. Without this comparison, it remains unclear whether the AI provides any diagnostic advantage over established clinical methods.
The reviewer’s valuable and insightful comment regarding the need to benchmark the AI model against expert assessment or conventional radiomorphometric indices, such as the mandibular cortical index, is greatly appreciated.
As the study had already been completed, it was not possible to modify the methodology to include such comparisons, since they were not part of the original retrospective design.
However, this important limitation has been explicitly acknowledged in the revised Discussion section. The following statement has been added:
“Another limitation is the lack of comparison between the performance of AI models and that of expert radiologists or conventional radiomorphometric indices such as the mandibular cortical index. While these comparisons could be important for evaluating clinical utility, they were not included in the present retrospective design and are suggested as a key objective for prospective validation studies.”
This addition aims to transparently recognize the absence of direct benchmarking and to emphasize it as a priority for future research. The authors are grateful for this constructive recommendation, which contributed to improving the completeness of the manuscript.

Round 2
Reviewer 4 Report
Comments and Suggestions for Authors
I have reviewed the revised manuscript entitled "Development of AI-Based Predictive Models for Osteoporosis Diagnosis in Postmenopausal Women from Panoramic Radiographs." The authors have made significant improvements, and the manuscript now presents a comprehensive and promising study on the application of artificial intelligence for osteoporosis screening in dental imaging. Below are my comments and suggestions for further improvement:
Major Strengths:
Substantial methodological enhancement: The revised version demonstrates a more structured comparison between classical radiomics, deep radiomics, and end-to-end CNN approaches. The external validation cohort enhances the study’s generalizability.
Comprehensive results presentation: The manuscript now clearly distinguishes the performances of different models, with appropriate evaluation metrics (AUC, accuracy, sensitivity, specificity) and detailed tables.
Expanded discussion: The authors have added valuable discussion points, including model overfitting, limitations of manual ROI selection, and clinical relevance. These additions increase the robustness of the interpretation.
Suggestions for Improvement:
Language and grammatical clarity:
Several sentences throughout the manuscript still suffer from awkward phrasing, grammatical errors, or unclear constructions — especially in the abstract and introduction. A professional English-language editing service or native speaker review is strongly recommended.
Example issues:
“The90.5%. Overall, while internal testing…” – this is syntactically incorrect and difficult to follow.
Repetitive or unclear expressions such as “The best performance was achieved by... applied on features extracted using…” could be streamlined.
Simplify and streamline tables:
The results tables (especially for deep radiomics and CNNs) are dense and repetitive. Consider merging similar rows or summarizing key findings in the main text, while moving extensive numerical comparisons to the Supplementary Materials.
Clarify overfitting concerns:
Some models show extremely high performance on internal datasets (e.g., AUC = 1.0), which is unlikely in realistic clinical scenarios. While this is partially addressed in the discussion, the authors should more explicitly acknowledge potential overfitting and its implications for clinical deployment.
Missing comparator (classical indices / expert performance):
While the study excludes radiomorphometric indices to avoid bias, a discussion or comparison with existing radiographic scoring systems (e.g., Mandibular Cortical Index) or human expert performance would further enhance the clinical context of the findings.
Final Recommendation:
Minor Revision.
The manuscript is scientifically sound and represents a relevant contribution to the field.
Author Response
We would like to thank you for your thorough and insightful review of our manuscript. We greatly appreciate your positive assessment and constructive suggestions, which have been instrumental in improving the overall quality and clarity of our work.
In response to your valuable feedback, we have carefully revised the manuscript. We focused on enhancing the linguistic accuracy throughout the text, streamlining the presentation of results by simplifying tables and relocating detailed analyses to the Supplementary Materials, and expanding the Discussion section to better address key aspects such as overfitting concerns and the comparison with conventional radiographic indices and expert performance.
Below, we provide detailed responses to each of your comments and describe the specific changes made to the manuscript.
I have reviewed the revised manuscript entitled "Development of AI-Based Predictive Models for Osteoporosis Diagnosis in Postmenopausal Women from Panoramic Radiographs." The authors have made significant improvements, and the manuscript now presents a comprehensive and promising study on the application of artificial intelligence for osteoporosis screening in dental imaging. Below are my comments and suggestions for further improvement:
Major Strengths:
Substantial methodological enhancement: The revised version demonstrates a more structured comparison between classical radiomics, deep radiomics, and end-to-end CNN approaches. The external validation cohort enhances the study’s generalizability.
Comprehensive results presentation: The manuscript now clearly distinguishes the performances of different models, with appropriate evaluation metrics (AUC, accuracy, sensitivity, specificity) and detailed tables.
Expanded discussion: The authors have added valuable discussion points, including model overfitting, limitations of manual ROI selection, and clinical relevance. These additions increase the robustness of the interpretation.
We sincerely thank you for these encouraging comments and for recognizing the efforts we made to improve the manuscript. Your detailed evaluation has been extremely valuable in helping us refine and strengthen our work.
Suggestions for Improvement:
Language and grammatical clarity:
Several sentences throughout the manuscript still suffer from awkward phrasing, grammatical errors, or unclear constructions — especially in the abstract and introduction. A professional English-language editing service or native speaker review is strongly recommended.
Example issues:
“The90.5%. Overall, while internal testing…” – this is syntactically incorrect and difficult to follow.
Repetitive or unclear expressions such as “The best performance was achieved by... applied on features extracted using…” could be streamlined.
Thank you for highlighting the issues related to language and grammatical clarity. In response to your valuable feedback, the entire manuscript has been carefully revised to improve linguistic accuracy, readability, and scientific clarity. All sections, including the Abstract and Introduction, have been reviewed and refined to eliminate awkward phrasing and enhance fluency and formal style.
We believe that these comprehensive revisions have substantially improved the overall quality and clarity of the manuscript.
Simplify and streamline tables:
The results tables (especially for deep radiomics and CNNs) are dense and repetitive. Consider merging similar rows or summarizing key findings in the main text, while moving extensive numerical comparisons to the Supplementary Materials.
Thank you for your suggestion. We have streamlined the presentation of the results by summarizing the best-performing model for each CNN architecture in Table 4. The table now reports both the performance metrics and their 95% confidence intervals for a more rigorous statistical interpretation. The full set of detailed results for all transfer learning strategies has been moved to the Supplementary Materials.
Similarly, Table 3 has been simplified to report only the best-performing model (Logistic Regression) for each deep radiomics feature set extracted from the different CNN architectures. This approach focuses the reader's attention on the key findings without redundant information, while complete numerical results for all models and feature sets are provided in the Supplementary Materials. These modifications aim to enhance readability and ensure that the main text highlights the most clinically relevant outcomes.
Clarify overfitting concerns:
Some models show extremely high performance on internal datasets (e.g., AUC = 1.0), which is unlikely in realistic clinical scenarios. While this is partially addressed in the discussion, the authors should more explicitly acknowledge potential overfitting and its implications for clinical deployment.
We thank the reviewer for this observation. We agree that the extremely high performance observed on internal datasets likely reflects overfitting, a well-known limitation in AI models, especially when dealing with relatively small sample sizes. To address this, we have revised the Discussion section to more explicitly acknowledge the potential for overfitting and its implications for clinical translation. Specifically, we now state:
"Nonetheless, the relatively small sample size may still pose a risk of overfitting, particularly for more complex CNN architectures. This limitation could compromise the generalizability of the models to new populations and different imaging conditions. Although internal testing yielded high AUC values, including instances of apparent perfect classification (AUC = 1.0), such results are likely indicative of overfitting to the training data rather than true model performance. The more modest outcomes observed during external validation reinforce the need for cautious interpretation and highlight the importance of rigorously assessing generalizability. Overfitting remains a significant concern, especially when working with limited datasets, and must be carefully addressed through the development of larger, prospective multicenter studies before these AI models can be reliably deployed in clinical practice. As a first step to mitigate this risk, we included an independent external validation cohort in our study; however, further large-scale validation remains necessary."
We believe this revision clarifies the issue and highlights both the limitations of the current study and the necessary future steps to ensure clinical applicability.
Missing comparator (classical indices / expert performance):
While the study excludes radiomorphometric indices to avoid bias, a discussion or comparison with existing radiographic scoring systems (e.g., Mandibular Cortical Index) or human expert performance would further enhance the clinical context of the findings.
In response, we have expanded the Discussion section to better address the clinical context by including a discussion on the performance of conventional radiomorphometric indices and human expert assessments. Specifically, we have added the following text:
"In particular, studies have demonstrated that radiomorphometric indices such as the Mental Index (MI), Gonial Index (GI), and Antegonial Index (AI) extracted from OPTs can be moderately accurate in identifying patients at risk of low bone mineral density. Additionally, combining these indices with patient factors like BMI and age has been explored to enhance predictive power, although with limited sensitivity. Nakamoto et al. (2003) also investigated the diagnostic performance of untrained general dental practitioners in visually assessing mandibular cortical erosion from OPTs, revealing that panoramic radiographs could be a useful adjunct in identifying postmenopausal women who should be referred for DXA."
Furthermore, we have explicitly acknowledged the absence of a direct comparison between our AI models and human experts or classical radiomorphometric indices as a limitation of the present study, by adding:
"Another limitation is the lack of comparison between the performance of AI models and that of expert radiologists or conventional radiomorphometric indices such as the mandibular cortical index. While these comparisons could be important for evaluating clinical utility, they were not included in the present retrospective design and are suggested as a key objective for prospective validation studies."
We believe these additions enhance the clinical relevance and completeness of the manuscript, as suggested by the reviewer.
